# A UV/Vis Spectroscopy-Based Assay for Monitoring of Transformations Between Nucleosides and Nucleobases

**DOI:** 10.3390/mps2030060

**Published:** 2019-07-15

**Authors:** Felix Kaspar, Robert T. Giessmann, Niels Krausch, Peter Neubauer, Anke Wagner, Matthias Gimpel

**Affiliations:** 1Bioprocess Engineering, Department of Biotechnology, Technische Universität Berlin, Ackerstraße 76, ACK24, D-13355 Berlin, Germany; 2BioNukleo GmbH, Ackerstraße 76, D-13355 Berlin, Germany

**Keywords:** UV/Vis spectroscopy, spectral unmixing, discontinuous assay, enzymatic reaction monitoring, nucleosides, nucleobases

## Abstract

Efficient reaction monitoring is crucial for data acquisition in kinetic and mechanistic studies. However, for conversions of nucleosides to their corresponding nucleobases, as observed in enzymatically catalyzed nucleoside phosphorylation reactions, the current analytical arsenal does not meet modern requirements regarding cost, speed of analysis and high throughput. Herein, we present a UV/Vis spectroscopy-based assay employing an algorithm for spectral unmixing in a 96-well plate format. The algorithm relies on fitting of reference spectra of nucleosides and their bases to experimental spectra and allows determination of nucleoside/nucleobase ratios in solution with high precision. The experimental procedure includes appropriate dilution of a sample into aqueous alkaline solution, transfer to a multi-well plate, measurement of a UV/Vis spectrum and subsequent in silico spectral unmixing. This enables data collection in a high-throughput fashion and reduces costs compared to state-of-the-art HPLC analyses by approximately 5-fold while being 20-fold faster and offering comparable precision. Additionally, the method is robust regarding dilution and sample transfer errors as it only considers spectral form and not absolute intensity. It can be applied to all natural nucleosides and nucleobases and even unnatural ones as demonstrated by several examples.

## 1. Introduction

Nucleoside modifying enzymes such as nucleoside phosphorylases are broadly applied for the synthesis of nucleoside analogs and pentose-1-phosphates [1,2,3]. Nucleoside phosphorylases catalyze the reversible phosphorolytic cleavage of the nucleobase from the nucleoside while facilitating formation of the corresponding pentose-1-phosphate (Scheme 1).

However, current approaches to the monitoring of these reactions have inherent drawbacks. Conventional assays are usually HPLC-based and require considerable experimental effort in sample preparation, financial investment into equipment and running costs, and long analysis times. UV/Vis-spectroscopy-based assays have also been employed for the monitoring of nucleoside phosphorylation reactions in the past and have often relied on the detection of change in extinction at a specific wavelength [4,5]. Downfalls of these approaches typically include dilution errors, problematic signal-to-noise ratios and a very limited range of suitable substrates.

Previously, we extended an existing assay for 2’-deoxythymidine phosphorylation reactions by normalization to the isosbestic point [6]. This enabled us to determine 2’-deoxythymidine/thymine ratios from aqueous mixtures and thus collect a large dataset with low experimental effort. Therein, the normalization allowed for correction of dilution errors and straightforward transfer of the assay to multi-well plates, which facilitated high-throughput experimentation.

In revisiting our data set, we envisioned that it should be possible to extend the recognition of the conversion ratio from a dual-wavelength observation to consideration of the full spectra and thus expand the scope of possible substrates and improve accuracy. Related concepts are known in the literature as hyperspectral imaging or spectral unmixing and are employed widely in fields such as remote sensing, fluorescence microscopy or in the food industry [7,8,9,10]. Following the literature terminology, we herein refer to our approach as spectral unmixing. Spectral unmixing relies on the concept of a linear combination of absorption spectra. When the spectra of pure compounds are known, any mixture can be generated from the pure spectra by addition and multiplication. Vice versa, any spectrum of a mixture of two compounds with known reference spectra can be traced back to a ratio of its individual constituents [11].

In this study, we systematically investigated the absorption spectra and pH-dependent spectral shift of all natural nucleosides and their bases and several examples of modified nucleosides, established suitable working conditions and implemented a spectral unmixing algorithm for data analysis. We demonstrate the accuracy and broad scope of our method, present its application to several enzymatic nucleoside phosphorylations and provide the freely available implementation of our Python algorithm for spectral unmixing and data treatment [12].

## 2. Materials and Methods

### 2.1. Chemicals

All nucleosides and nucleobases in this study were purchased from Sigma Aldrich, TCI or Carbosynth. All other chemicals were obtained from Carl Roth at the highest available quality. Water used was deionized to 18.2 MΩ∙cm with a Werner water purification system. NaOH solutions were prepared with deionized water.

### 2.2. Enzymes

The enzymes used in this study were pyrimidine nucleoside phosphorylase (Py-NPase, EC 2.4.2.2, NCBI sequence accession number WP_041270053.1) and purine nucleoside phosphorylase (Pu-NPase, EC 2.4.2.1, NCBI sequence accession number WP_064551770.1) from *Bacillus thermoglucosidasius* (DSM No.: 2542). The *N*-terminally His_6_-tagged enzymes were purified from *Escherichia coli* BL21 following IPTG-induced overexpression. Purification was achieved via Ni-NTA affinity chromatography as described previously [13,14] and purity was assessed by SDS-PAGE and determined to be > 90%. Afterwards, the enzymes were desalted against 2 mM KH_2_PO_4_ buffer (pH 7, 25 °C) by an ÄKTA system with a pre-equilibrated HiPrep 26/10 Desalting Column, and stored until use at 4 °C at concentrations of 1.61 and 6.63 mg/mL, respectively, as judged by NanoDrop analysis (calculated with 1 AU at 280 nm = 1 mg/mL).

### 2.3. Recording of Spectra

Spectra were recorded from 250 to 350 nm in steps of 1 nm on a BioTek PowerWave HT plate reader using UV/Vis-transparent 96-well plates (UV-STAR F-Bottom #655801, Greiner Bio-One). The measurement was generally performed directly after the experiment, but overnight storage of diluted alkaline samples in sealed Eppendorf tubes did not impede results. Sample storage in the 96-well plate is not recommended as we generally observed issues such as solvent evaporation causing sample precipitation after several hours already.

### 2.4. Investigating the pH Dependence of Spectra

The pH dependence of the UV/Vis absorption spectra of all nucleosides and nucleobases in this study was investigated between pH 7 and 13 in steps of 1 pH unit. The experiment was conducted with a roughly 100 µM (for pyrimidine nucleosides) or 80 µM (for purine nucleosides) solution of the respective compound in 200 mM K_2_HPO_4_ as pH-buffer. The pH was adjusted with NaOH and HCl to the desired pH value, and aliquots of 200 µL were taken from the solution and transferred directly into a 96-well plate for measurement. K_2_HPO_4_ buffer was previously determined to have no effect on the UV absorption spectra of any of the nucleosides investigated here.

### 2.5. Enzymatic Reactions

Enzymatic reactions were prepared from stock solutions of nucleoside substrate, buffer, phosphate and water and preheated to the reaction temperature before the reaction was started by the addition of enzyme solution (2–10 µL). Reactions were typically performed with 2 mM substrate and 10 mM phosphate in 50 mM glycine-NaOH buffer at pH 9.0 (measured at 25 °C) in a total volume of 500 µL. Temperature and enzyme concentration were adjusted according to the requirements of the experiment.

### 2.6. Enzyme Reaction Sampling

Enzyme reaction sampling was conducted as described recently [6]. Briefly, from reactions with 2 mM substrate, 30 µL (for pyrimidine substrates) or 20 µL (for purine substrates) of sample were drawn and pipetted into NaOH (generally 100 mM, exceptions are mentioned in Table 1) in a separate Eppendorf tube (final volume of 500 µL). Quenching of the enzyme reaction was achieved via brief shaking of the sample tube. Subsequently, 200 µL of the diluted sample was transferred into a well of a 96-well plate for measurement. Whenever different substrate concentrations were used, sample and NaOH volumes were adjusted accordingly to yield the same final concentration of UV-active compounds.

### 2.7. Spectral Unmixing

Spectral unmixing was performed by the formula:(1)$$x=argmin_{x}\left({\left(Y_{k}-\overset{M}{\underset{i=1}{\sum }}x_{i}Y_{i}\right)}^{2}\right),$$
with:
x = representation of the molar fractions of compounds for sample k in a vector form $x=\left[x_{1},x_{2},\ldots \;,x_{M-1},\;x_{M}\right]$ (in this study: $x=\left[x_{1},x_{2}\right]$)$Y_{k}$ = experimentally determined spectrum for sample k,M = number of compounds (in this study: M=2),$x_{i}$ = molar fraction of compound i, and$Y_{i}$ = spectra of pure compound i, normalized to the corresponding isosbestic point of the mixture (see below),
where $argmin_{x}\;f\left(x\right)$ gives the molar fractions x for which f(x) yields the smallest value (i.e., f(x) is minimized).

The molar fractions of each compound were restricted to real-world solutions, i.e., 0% to 100%, by applying the bounds:(2)0≤x≤1
and demanding that the individual molar fractions must sum to 100%, i.e., applying the constraint:(3)$$\overset{M}{\underset{i=1}{\sum }}x_{i}=1$$

The minimization was conducted by the least_squares implementation of the freely available Python package, lmfit [15]. Similarly, the standard errors of the estimated molar fractions were estimated by lmfit. If not stated differently, fitting was performed from 250 nm to the respective compound-specific upper limit listed in Table 1. The developed Python software for data treatment and analysis is available online [12].

### 2.8. Normalization of Spectra to Accommodate Concentration Differences

For a given nucleoside–nucleobase pair, there usually exists an isosbestic point at $\lambda_{i,j}$, at which the absorbance stays constant throughout all arbitrary mixtures of the pure compounds i and j (see title graphic or [6] for visualization). The blank-corrected reference spectra and the blank-corrected spectrum to fit were normalized to this isosbestic point $\lambda_{i,j}$, via division by the absorption at this wavelength:(4)$$Y_{i}=\frac{\mu_{raw,i}-\mu_{blank}}{Abs{\left(\mu_{raw,i}-\mu_{blank}\right)}_{\lambda_{i,j}}}$$
with:
$Y_{i}$ = the normalized spectrum of compound i,$\mu_{raw,i}$ = raw spectrum of compound i,$\mu_{blank}$ = raw blank spectrum of NaOH solution,$Abs{\left(\mu_{raw,i}-\mu_{blank}\right)}_{\lambda_{i,j}}$ = the absorption of the blank-corrected spectrum at wavelength $\lambda_{i,j}$.

### 2.9. Statistical Analyses

The coefficient of determination (R^2^) for analysis of actual vs. predicted data (Section 3.3) was calculated as follows:(5)$$R^{2}=1-\frac{SS_{res}}{SS_{tot}}$$
where:
$SS_{res}$ = residual sum of squares of all data points;$SS_{tot}$ = total sum of squares of all data points;
with:(6)$$SS_{res}=\overset{n}{\underset{i=1}{\sum }}{\left(y_{i}-\bar{x}\right)}^{2}$$
(7)$$SS_{tot}=\overset{n}{\underset{i=1}{\sum }}{\left(y_{i}-x_{i}\right)}^{2}$$
with:
$y_{i}$ = predicted molar fraction of the data point i of the data set (i.e., all data points for one compounds, considering both replicates), in percent,$\bar{x}$ = average of all actual molar fraction values of the data set (equal to 50%), in percent,$x_{i}$ = actual molar fraction value for the data point i, in percent,n = total number of data points in the data set,
where “actual” refers to the intended (designed) molar fraction in the sample and “predicted” refers to the molar fraction as estimated by the algorithm.

The root mean square deviation (RMSD) for evaluation of fine-tuning of the algorithm (Section 3.5) was calculated as follows:(8)$$RMSD=\sqrt{\frac{\sum_{i=1}^{n}{\left(y_{i}-x_{i}\right)}^{2}}{n}}$$
with definitions as above.

Both methodologies can be inspected as openly available Python code from the Appendix A and at the external online repository [16].

### 2.10. HPLC Analysis of Samples

HPLC analysis of samples was performed with an Agilent 1200 Series HPLC system, employing a Phenomenex Kinetex Evo C18 100 Å column (250 mm × 4.6 mm) running a linear gradient of 3–40% acetonitrile in 20 mM NH_4_Ac over 10 min, as described previously [14]. Detected substances were matched with reference compounds and quantification was achieved via previously recorded calibrations.

## 3. Results

### 3.1. Nucleoside–Nucleobase Pairs Show Discriminable UV/Vis Spectra under Alkaline Conditions

Under neutral pH conditions, the spectra of most nucleosides and those of their free nucleobases show only insignificant differences in shape and absorption maximum. However, their spectral forms change with varying pH (Figure 1 and Appendix A).

For the exemplary 2’-deoxythymidine/thymine pair, we investigated the pH-dependent change of spectral properties by subjecting solutions of pure 2’-deoxythymidine and thymine to UV/Vis spectroscopy at pH values ranging from 7.0 to 13.0 (Figure 1a,b), as these are the feasible limits of the reaction dictated by the stability of the generated pentose-1-phosphates [6,17]. Although the absorbance of 2’-deoxythymidine generally decreases with increasing pH, the overall shape of its spectrum and its absorption maximum at 266 nm are not affected (Figure 1a). In contrast, the spectrum of thymine displays a characteristic shift towards increased absorption at higher wavelengths as it broadens significantly and its maximum is shifted to 287 nm (Figure 1b).

A similar behavior was also observed for all other investigated nucleoside–nucleobase pairs (Appendix A). The spectral shape, pH needed for deprotonation-induced spectral shift and the pH range of stable spectra varied significantly between nucleosides. Consequently, the pH value for analysis of certain nucleosides had to be adjusted to accommodate their behavior in alkaline solution. The spectral properties of all nucleosides in this study are given in Table 1. Interestingly, we found slight but persistent differences between the spectra of ribo- and 2’-deoxyribonucleosides regarding spectral shape and $\lambda_{max}$ (Table 1).

We observed different absorption spectra for each nucleoside and its corresponding base in alkaline solution. This led us to the conclusion that the differences in individual absorption spectra under alkaline conditions are sufficient to discriminate between nucleoside and free nucleobase in a mixture via spectral unmixing.

### 3.2. Spectral Unmixing Can Resolve the Composition of Nucleoside–Nucleobase Mixtures

We further investigated the prime precondition for linear spectra unmixing, the fact that mixtures (based on molarity, i.e., molar fractions) yield the corresponding linear combination of the spectra of their individual constituents, by measuring the pure spectra of 2’-deoxythymidine and thymine as well as an equimolar mixture of both compounds at pH 13 (Figure 1c). The spectrum of the 1:1 mixture shows an intermediate shape representing the average of both spectra via linear combination. To confirm our hypothesis, we further analyzed mixtures of 2’-deoxythmidine and thymine in a 9:1 (Figure 1d) and 1:9 (Figure 1e) ratios, respectively. Similarly, these mixtures could be decomposed with the linearly combined reference spectra of the pure compounds (data not shown).

### 3.3. The Accuracy of Spectral Unmixing Holds up Excellently across All Possible Mixture Compositions

To prove the accuracy of our methodology, we recorded spectra for six different mixtures (0–100% molar fractions, in 20% steps) of three nucleoside–nucleobase pairs, including one natural and one unnatural pyrimidine as well as one purine (Figure 2).

The results demonstrate that the spectral unmixing algorithm resolves mixtures of nucleosides and their bases with excellent accuracy (R² > 0.99). Conveniently, this approach inherently adjusts for experimental errors such as pipetting inaccuracies during alkaline dilution or transfer to the microtiter plate that affect the optical pathlength, since the results only depend on the spectral form and not the absolute intensity. Furthermore, by working under strongly basic conditions, one can bypass common issues such as poor solubility of nucleosides and nucleobases, that are known to be especially problematic for purines (guanosine/guanine, inosine/hypoxanthine) and halogenated nucleosides (5-fluorouridine/5-fluorouracil).

Additionally, we also compared our method to state-of-the-art HPLC analysis. Therefore, we subjected the 2’-deoxythymidine/thymine mixtures used for the generation of the dataset from above to HPLC. The results of both methods were in excellent agreement and differed less than 1.5 percentage points from each other (Table 2).

### 3.4. Spectral Unmixing Empowers Practical Applications in Enzymatic Conversion Research

With this versatile tool in hand, we monitored the reaction progress of a number of nucleoside phosphorylation reactions with different substrates and enzymes, showcasing its broad applicability (Figure 3 and Appendix A).

As shown in Figure 3, the assay allowed the observation of the reaction progress from initial rate kinetics towards the equilibrium. Previous recording of reference spectra for the substrate and product of the reaction enabled direct application of the method and quick data acquisition and analysis. Similar results were obtained for the phosphorolytic cleavage of the purine nucleosides 2’-deoxyadenosine and 2’-deoxyinosine (Appendix A). It is noteworthy, that the exact molar extinction coefficients did not need to be determined at any time, as normalization to the isosbestic point of reference spectra and experimental spectra corrected for them. These examples demonstrate the capabilities of our method for straightforward monitoring of enzymatic nucleoside phosphorylation reactions with natural and modified substrates.

### 3.5. The Spectral Unmixing Algorithm Can be Fine-Tuned to Specific Areas of Interest

When investigating mixtures in extremely high or low composition regimes (for example < 5% molar fraction of the nucleobase) we could typically observe slight deviations of < 0.5 percentage points from actual to predicted values (Figure 4). To identify the source of error in a structured manner, we herein adhere to the ISO definition of accuracy (“trueness”) and precision (“cloneness”) of the result.

Thus, we first investigated precision of the measurements by the specific used equipment. Successive measurements of the same multi-well plate yielded deviations in the range of 0.3 percentage points, an error which cannot be influenced by our algorithm but is determined by the type of equipment, in our case a plate reader for UV/Vis spectroscopy. Successive calls of the algorithm on the same data performed with neglectable standard errors (<10^−5^ percentage points; data not shown). This underlines the high precision of our algorithm but points to limitations of the instrumentation which have to be addressed by the individual researcher.

Secondly, to improve the accuracy of the algorithm’s predictions in this region, we selected only a part of the full recorded spectra for fitting. In this case, we limited the region for evaluation to the range of 275–310 nm, which comprises the information-rich region starting from the isosbestic point. This increased the accuracy in this composition range by almost 50% as judged from the RMSD (full spectrum: RMSD = 0.52 pp, 275–300 nm: RMSD = 0.26 pp, considering replicate 1 and 2; Figure 4).

## 4. Discussion

In this contribution, we show that spectral unmixing is a feasible method for monitoring base cleavage reactions with natural and modified nucleosides.

### 4.1. The Prerequisites of Spectral Unmixing are Met by All Nucleoside–Nucleobase Pairs

The prerequisites of spectral unmixing are (i) absorption in the UV/Vis region, (ii) differentiable UV/Vis spectra of nucleoside and nucleobase in alkaline solutions, (iii) stability of the compounds in alkaline solution and (iv) linear combination of their individual spectra in mixture. All of these were met universally across the investigated nucleosides in the present study (Figure 1, Figure 2 and Appendix A). This shows that our methodology can be generalized and will likely be applicable to almost all nucleosides and nucleobases.

Special cases for which prerequisites (i) and (ii) may not be fulfilled are non-aromatic nucleobases such as 5,6-reduced pyrimidines or base in which tautomerization is hindered by *N*-alkylation. This is because the spectral shift of the free nucleobase under basic conditions originates from its deprotonation and tautomerization to an aromatic system [18]. The nucleobase protons generally have a pK_a_ < 12 due to their partial phenolic character and are thus easily abstracted in alkaline solution [19]. Some modified nucleobases, such as 5-halogenated uracils, even display two deprotonation steps between pH 7 and 13, which are enabled by the electron-withdrawing substituent. On the other hand, N1-substitution (such as in pyrimidine nucleosides) inhibits the tautomerization of the base across the N1 position, and thus increases the pK_a_ value of the amine protons outside the limits of aqueous solutions for nucleosides (pKa > 14). Thus, the UV absorption properties of nucleobases are typically indistinct from their corresponding nucleosides at natural pH, while a characteristic spectral shift of the nucleobase occurs under strongly basic conditions (pH > 12). This trend has been reported by early studies in the field [18,20,21] and was confirmed and exploited in this paper.

Regarding prerequisite (iii), all nucleosides in this study were found to be stable at high pH values, enabling measurement in alkaline solution. However, this may not be the case for all nucleosides. Particularly, instable nucleosides such as 7-methylguanosine, 8-oxoguanosine or 2-fluoroadenosine might not be stable enough over sufficiently long time periods to enable reliable measurement under the proposed conditions. Thus, nucleosides not included in this study may need to be evaluated on a case-to-case basis.

Finally, although spectral unmixing yields a higher precision at very alkaline pH, we want to stress that analysis of nucleoside/nucleobase mixtures via spectral unmixing is principally possible over the whole pH range, depending on the application, substrates and required precision. This opens further routes for direct and continuous measurements of biocatalytic reactions.The concept of spectral unmixing is introduced into UV/Vis spectroscopy of nucleoside/nucleobase mixtures.

To the best of our knowledge, the concept of spectral unmixing has not been applied in UV/Vis spectroscopy of nucleoside/nucleobase mixtures previously. Related concepts are, however, known in the field as shoulder correction or deconvolution of spectra, both of which refer to least squares curve fitting [18,22]. In other areas of biology and chemistry, spectral unmixing is being applied already, such as in flow cytometry, chemometrics and chromatography [23,24,25]. Compared to one- or two-wavelength detection methods, the data generated with our proposed methodology demands increased computational effort, which can be handled by a conventional personal computer, and presents a challenge in terms of data organization, for which we present a possible data management scheme in the Appendix A.

### 4.2. Previous Assays in the Field

Although multi-variate/multi-dimensional methodologies are seldomly applied in UV/Vis spectroscopy, there are multiple examples of assays which have previously been applied in this field. For example, UV-spectroscopy has recently been employed to qualitatively monitor enzymatic ribosylation reactions of artificial nucleobases [26].

Beyond that there are several other non-HPLC-based assays for the monitoring of nucleoside phosphorylations found in the literature [27]. In general, these are either (i) based on fluorescence of selected artificial nucleobases and their corresponding nucleosides, such as 4-thiopyrimidines [28], 8-azapurines [29,30,31] or tricyclic ribosides [26,29,32], or based on the only natural fluorescent nucleobase guanine [33]; (ii) coupled assays employing the in situ coupling of a reaction of interest with the oxidation of hypoxanthine to uric acid (monitored UV/Vis spectrophotometrically at 293 nm) via xanthine oxidase [34] or with pyrophosphatase (monitored colorimetrically with malachite green) [35]; or (iii) one-wavelength detection methods for uridine and thymidine [4,5] or guanosine [36].

All of these assays are limited in regard to the chemical space that can be probed with them, as they are only applicable to a handful of nucleosides and thus, are very specific. For universal assaying, HPLC is necessary and consequently, widely used in the field. Our method, on the other hand, can be used for all natural and a wide range of modified nucleosides and thus relieves the constraints of existing assays regarding substrate selection.

### 4.3. Improvements and Adaptations of the Assay

We found our algorithm to be precise, yet flexible enough to allow investigations across the full range of possible mixtures of any nucleoside and its free base (and thus for monitoring the full reaction progress of nucleoside base cleavages).

Overall, our method is reliable across the full range of mixtures with errors in the range of 1–2 percentage points when using the full spectrum and single measurements. In the case of 2’-deoxythymidine and thymine (Figure 4), accuracy at low base/nucleoside ratios could be improved by selecting a certain wavelength range for fitting. Nonetheless, differences in the 0.3 percentage point range remained between measurements due to our specific experimental instrumentation and its inherent inaccuracy, even with excellent signal-to-noise ratios in the range of 50–100 (e.g., signals of 0.394 ± 0.005 AU over a blank of 0.032 ± 0.005 AU at the isosbestic point of 2’-deoxythymidine base cleavage at 278 nm). If precision beyond this is required, it could thus be recommendable to employ repeated measurements or adapt the instrumental setup. Conveniently, the necessary optimization and validation can be automatized with our methodology by employing liquid handling systems to generate mixtures of interest and mathematical algorithms for selection of optimal wavelength regimes.

## 5. Conclusions

Herein, we provide a highly valuable tool for all researchers working in the field of nucleoside–nucleobase conversions. These conversions can be followed whether they are being catalyzed chemically or enzymatically, and are independent of whether they are hydrolytic, phosphorolytic or other processes. Our approach eliminates the need for HPLC in one-step base cleavage reactions by a 20-fold reduction of data acquisition time and a 5-fold decrease in cost while offering comparable accuracy. The proposed method allows high experimental throughput and ultimately, encourages scientists to broadly probe their enzymes’ working space. Thus, effects of reaction pH, temperature, enzyme and substrate concentrations, as well as the influence of impurities or other reagents can easily be investigated, using only minimal resources and time.

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
