# Peer review of "A UV/Vis Spectroscopy-Based Assay for Monitoring of Transformations Between Nucleosides and Nucleobases"

_mps, 2019, doi:10.3390/mps2030060_

Round 1

Reviewer 1 Report

In the article entitled "A UV/Vis Spectroscopy-Based Assay for Monitoring of Transformations Between Nucleosides and Nucleobases" authors propose the simple and fast assay to determine the relative quantity of nucleosides and corresponded nucleobases. The method based on the significant difference of the UV bands of the considered compounds especially in strongly alkaline solutions.  The analysis is quite simple and it is surprisingly that nobody used it before at least in titled area.

The key moment is dramatic changing of the bands of nucleobases upon the varying of media pH, while the bands of nucleosides remain approximately the same. I believe that it can be matter of the additional study. For example, the band changing is connected only with pH or is it the result of an interaction of the nucleobase, let say, with NaOH? This, of course, does not influence on the results of the assay.

The article is written clearly, the results can be interesting for researchers studying the reactions between nucleosides and nucleobases, and the article can be accepted after the minor changes.

1.      Page 3, formula (1). There are no definition of ω and argmin(ω) function. I believe that argmin(ω) came from some computer language and return the array of parameter values, where the function in parentheses reaches the minimum value. However in other languages this function returns the index of array with the minimal value. Moreover I sure that the major of reader do not know about this function. So, define please argmin(ω) and ω

2.      Page 4, formula (2). Probably you mean ωi. Or do you mean that 0 and 1 are arrays?

3.      Formulas (6), (7), (8). Here you reintroduce the x and y. It slightly impedes understanding of the article here. Please consider using another letters for molar fractions or for the spectra in formula (1).

4.      Page 6, lines 195 – 197. "Interestingly, we found slight but significant differences between the spectra of ribo- and 2’-deoxyribonucleosides regarding spectral shape and ???? (Table 1)." This difference does not exceed 2 nm and in most cases is 1 nm. According the PowerWave Operator's Manual, the λ accuracy is ±2 nm and λ bandpass is 5 nm. So I would say that ???? for ribo- and 2’-deoxyribonucleosides are the same within an experimental error.

5.      "The spectral unmixing algorithm can be fine-tuned to specific areas of interest" section. When you discuss the investigating mixtures in extremely high or low composition regimes it is also interesting to know the values of noises in UV spectra in the band region. Because it seems the most probable issue of deviation of predicted molar fractions from actuals ones. Consider please description of analysis of signal to noise ratio in UV spectra.

Reviewer 2 Report

The manuscript from Beverly, Hagen and Slack reports an UV-VIS based assay for nucleoside phosphorylase-catalyzed reactions. The assay relies on determination of nucleobase/nucleoside ratio by UV-VIS measurments in a 96-well plate reader followed by "UV-VIS spectra unmixing", which is based on the corresponding reference spectra and an algorithm developed by the authors. As the UV-VIS measurement has to be performed at pH ~13, the assay can be used only in discontinuous mode, which I guess is the main limitation of this approach. Nonetheless, compared to the current state-of-the-art approach, which is the HPLC method, the approach taken by the authors provides significant advantages without compromising the robustness (as shown experimentally and pointed out in the discussion). I like the fact that the authors took a comprehensive approach and demonstrated applicability of the assay to many different natural an unnatural deoxy- and ribonucleoside/nucleobase pairs. I also appreciate that the code with the algorithm has been made available to others. I anticipate that the results will be highly useful to many other researches in the field. The manuscript is very clearly written, the result are neatly presented and briefly yet soundly discussed. Thus, I have no major cristism. My only suggestion would be to include/mention one more possible limitation in the discussion part (page 10-11, paragraph entitled "the prerequisites.."). Namely, I assume that the assay would not be applicable to nucleosides that are chemically unstable under alkaline pH. As an example I can name 7-methylguanosine and 8-oxo-guanosine, but I guess other examples can be found. I hope that the authors agree that it would be useful to mention this potential problem to those interested in using the method.

In conclusion, I recommend publication of the manuscript after minor revision.
